# Influence of V on the Microstructure and Precipitation Behavior of High-Carbon Hardline Steel during Continuous Cooling

**DOI:** 10.3390/ma17061392

**Published:** 2024-03-19

**Authors:** Junxiang Zhang, Shangjun Gu, Jie Wang, Fulong Wei, Zhiying Li, Zeyun Zeng, Bin Shen, Changrong Li

**Affiliations:** 1College of Materials and Metallurgy, Guizhou University, Guiyang 550025, China; m1476776890@163.com (J.Z.); zzy2686412958@163.com (Z.Z.);; 2Guizhou Province Key Laboratory of Metallurgical and Process Energy Saving, Guiyang 550025, China; 3Shougang Shuicheng Iron and Steel (Group) Co., Ltd., Liupanshui 553000, China

**Keywords:** high-carbon hardline steels, microstructure, VC, two-dimensional misfit, carbon diffusion coefficient, PTT

## Abstract

High-carbon hardline steels are primarily used for the manufacture of tire beads for both automobiles and aircraft, and vanadium (V) microalloying is an important means of adjusting the microstructure of high-carbon hardline steels. Using scanning electron microscopy (SEM), X-ray diffraction (XRD), and transmission electron microscopy (TEM), the microstructure and precipitation phases of continuous cooled high-carbon steels were characterized, and the vanadium content, carbon diffusion coefficient, and critical precipitation temperature were calculated. The results showed that as the V content increased to 0.06 wt.%, the interlamellar spacing (ILS) of the pearlite in the experimental steel decreased to 0.110 μm, and the carbon diffusion coefficient in the experimental steel decreased to 0.98 × 10^−3^ cm^2^·s^−1^. The pearlite content in the experimental steel with 0.02 wt.% V reached its maximum at a cooling rate of 5 °C·s^−1^, and a small amount of bainite was observed in the experimental steel at a cooling rate of 10 °C·s^−1^. The precipitated phase was VC with a diameter of ~24.73 nm, and the misfit between ferrite and VC was 5.02%, forming a semi-coherent interface between the two. Atoms gradually adjust their positions to allow the growth of VC along the ferrite direction. As the V content increased to 0.06 wt.%, the precipitation-temperature-time curve (PTT) shifted to the left, and the critical nucleation temperature for homogeneous nucleation, grain boundary nucleation, and dislocation line nucleation increased from 570.6, 676.9, and 692.4 °C to 634.6, 748.5, and 755.5 °C, respectively.

## 1. Introduction

Pearlite is the predominant microstructure in high-carbon hard wire steels, with the size of pearlite colonies and the interlamellar spacing (ILS) being the main parameters determining the mechanical properties of steel [1,2]. Microalloying is a common method to adjust the microstructure of steel, control precipitation behavior, and enhance the mechanical properties of steel [3]. Researchers are dedicated to obtaining the optimal addition of V to control the size of pearlite colonies and ILS and to develop pearlitic steels with superior mechanical properties.

V refines grains primarily by precipitation but also by its presence in a solid solution, enhancing the mechanical properties of steel [4,5]. When V exists in a solid solution within the austenite, it can inhibit the recrystallization of austenite grains [6]. Studies by Tian et al. [7] have shown that V can drag carbon atom diffusion and reduce the growth rate of pearlite and the thickness of cementite. During the pearlite transformation, V is distributed into the cementite lamellae and replaces Fe atoms in cementite, enhancing stability, Young’s modulus and shear modulus of cementite, and reducing the elastic anisotropy of cementite [8,9]. However, excessive V can increase the content of proeutectoid ferrite at grain boundaries and reduce the plasticity of cementite lamellae, leading to the discontinuity of pearlite lamellae and promoting the formation of bainite and martensite [10]. The shape of VC can be spherical, discoidal, ellipsoidal, etc. [11,12]. When austenite transforms into ferrite, the decreasing solid solubility of V results in the substitution of V atoms for Fe atoms within the ferrite matrix, which results in the formation of an intermediate coherent crystal structure within the ferrite matrix [13]. As the particles grow, the precipitates gradually lose coherency and grow into disc-shaped or plate-shaped forms [13]. The precipitation of VC occurs simultaneously with the γ→α phase transformation, and after the phase transformation is complete, VC continues to precipitate [14]. The orientation relationship between the VC precipitates and the matrix is of the Baker–Nutting (B-N) type or its variants [15]. Moon et al. [16] investigated V–Nb heat-resistant steel and observed that during the γ to α transformation process, nano-sized (V, Nb) (C, N) particles precipitate along the migrating γ/α interface, displaying characteristics of interphase precipitation. The precipitated phase exhibits a B–N orientation relationship with the ferritic matrix, and the precipitated phase adopts a plate-like morphology. Studies by Bikmukhametov et al. [17] have shown that interphase precipitates form preferentially at K–S interfaces with a close-packed habit plane of (110)_α/_/(111)_γ_ and low mobility. The chemical composition of the precipitates is related to their size; smaller V carbides are rich in Fe, while larger precipitates contain Fe only near the matrix/precipitate interface [18]. The nano-precipitation of VC can pin the slip of dislocations during deformation, thus increasing the strength of the material [19]. When precipitates form at austenitic grain boundaries, they can also serve as nucleation sites for pearlite growth [20]. The precipitation behavior of VC is related to the phase transformation temperature, aging time, and cooling rate [21,22,23]. Lowering the phase transformation temperature can reduce the size of VC and increase its number density and hardness [21]. Increasing the aging time can increase the number density of precipitates, but it can also increase the size of precipitates [22]. Research by Zhang et al. [23] has shown that when the phase transformation temperature is reduced from 993 K to 923 K, the number density of plate-like precipitates at non-K–S (Kurdjumov–Sachs) interfaces significantly increases while further reducing the temperature to 873 K does not significantly change the number density of plate-like precipitates. Studies by Gu et al. [24] indicate that dislocations can serve as nucleation sites for nanoscale VN precipitates (<5 nm). With increasing aging time, the contribution of precipitation strengthening is partially counteracted by the softening attributed to dislocation recovery and bainite granularization. Singh et al. [25] demonstrated that composite precipitates formed by Nb and V in steel exhibit coarse sizes (70–80 nm) and low volume fractions (~0.5%). Consequently, the strengthening effect of (Nb, V) C precipitates on yield strength is limited (35 MPa). The combination of carbon with (Nb, V) C offers a key advantage in mitigating yield point elongation rather than enriching the ferrite (α) solid solution to achieve discontinuous yielding. Research by Maejima et al. [26] suggests that the precipitation of VC is not essential for the strengthening effect in pearlitic steels. V can alter the lattice parameters of cementite by dissolving into it, thereby inducing ferrite/cementite lattice mismatch, increasing lattice strain, and enhancing the strength of pearlitic steels.

There is not much data on the influence of V content on the microstructure and precipitation behavior of high-carbon hardline steels. This research is focused on the effects of vanadium content on the microstructure and precipitation behavior and elucidates the underlying mechanisms. This research investigates the variation in the volume fraction of pearlite and ILS with changes in vanadium content, as well as the properties of the VC precipitate phase interfaces. The influence of vanadium content on the microstructure, precipitate nucleation temperature, and time is explained using a carbon diffusion model and precipitation kinetics. This work can guide the microstructural adjustment and control of precipitate phases in V-microalloyed steels, providing a theoretical foundation and reference for the development and production of high-carbon hardline steels.

## 2. Materials and Methods

### 2.1. Materials Prepared

The experimental steel was melted in a high-temperature tube furnace in the laboratory. During the remelting process, vanadium iron alloy with different weights (0.44, 0.96, and 1.36 g) was added separately into a high-temperature tube furnace containing 0.7 wt.% C carbon structural steel, resulting in cast ingots of 1645.9, 1795.4, and 1692.2 g, respectively. The composition of the ingots was determined using an Inductively Coupled Plasma (ICP) spectrometer (PerkinElmer, Hopkinton, MA, USA), and the chemical compositions are presented in Table 1. The experimental steels with 0.02, 0.04, 0.06 wt.% V were named LV, MV, and HV steel, respectively.

### 2.2. Thermo-Mechanical Processing

The steel ingots were processed into cylindrical specimens of 8 mm × 12 mm and subjected to plane strain compression tests on the Gleeble 3800 thermal simulation machine (Data Sciences International, Albany, NY, USA). Based on solubility theory and related studies [27,28], the thermal simulation process for the experimental steel was designed as follows: the specimens were heated to 1200 °C at a rate of 20 °C·s^−1^ and held for 200 s to ensure the homogenization of alloying elements. Subsequently, they were cooled to 900 °C at 10 °C·s^−1^ and held for 20 s to eliminate temperature gradients, followed by single-pass hot deformation. The deformation temperature was 900 °C, with a true strain of 0.69 and a strain rate of 5 s^−1^. The deformed specimens were cooled to room temperature at 1, 2, 5, and 10 °C·s^−1^. The thermal simulation process flow is illustrated in Figure 1.

### 2.3. Characterization Methods

The center of the specimens under different cooling rates were etched using a 4% nitric acid ethanol solution. The microstructure of the experimental steels was characterized using SEM (SUPRA40, Carl Zeiss AG, Oberkochen, Germany). ILS and the proportions of ferrite and pearlite in the specimens were statistically analyzed. The phase compositions of the LV, MV, and HV steel samples were characterized using XRD (Rigaku Ultima IV, Rigaku, Akishima-shi, Japan) at a scanning speed of 5°/min. The foil thickness for TEM analysis was reduced to approximately 100 μm or less through mechanical polishing, then punched into discs of about 3 mm in diameter, followed by electrolytic polishing using a twin-jet electrolytic polisher (MTP-1A, Jiaoda Electro-mechanical, Shanghai, China) at a voltage of 30 V and a temperature of 15 °C. The electrolyte was composed of 15% perchloric acid by volume and 85% ethanol by volume. The morphology and composition of the microstructures and precipitates were characterized using TEM (Tecnai G2 F20 S-TWIN, FEI, Hillsboro, OR, USA) equipped with an Energy Dispersive Spectrometer (EDS). The raw high-resolution images (HRTEM) of the precipitates were processed using the HRTEM Filter function of GMS-3 (GATAN, Pleasanton, CA, USA). Fast Fourier Transform (FFT) and Inverse Fast Fourier Transform (IFFT) were applied to obtain the electron diffraction spots and lattice fringes of the matrix and precipitates. The electron diffraction spots were indexed, and the lattice spacings were measured to determine the corresponding crystallographic indices.

## 3. Results and Discussion

### 3.1. Microstructure

The microstructure and ILS of the three experimental steels are shown in Figure 2.

Figure 2 demonstrates that at cooling rates of 1, 2, and 5 °C·s^−1^, the microstructures of the three experimental steels consist of proeutectoid ferrite (PF) along grain boundaries and pearlite (P). With an increase in V content, the volume fraction of PF in the experimental steels gradually increases. Studies have shown [29] that V atoms at grain boundaries inhibit the diffusion of carbon atoms into the grain interior, leading to carbon segregation at the grain boundaries and, consequently, an increased tendency for ferrite precipitation, which results in an increased content of PF. As the cooling rate increases, the volume fraction of PF gradually decreases. At a cooling rate of 10 °C·s^−1^, all three experimental steels exhibit a blocky structure with serrated edges, which could potentially be bainite, martensite, or retained austenite. Figure 3 presents the Continuous Cooling Transformation (CCT) curves and XRD results of the experimental steels.

The CCT indicates that only pearlitic transformations occur in the experimental steels, with no martensitic phase transformations observed. XRD reveals the presence of solely ferritic phases in the steel, with no residual austenite detected, suggesting that the blocky structure observed may likely be bainite (B). Figure 3a–c demonstrate that increasing the V content or the cooling rate can delay the start temperature of the pearlitic transformation. As the cooling rate increases from 1 °C·s^−1^ to 10 °C·s^−1^, the start temperature of the pearlitic transformation in the LV, MV, and HV steels decreases by 93.1 °C, 66.5 °C, and 67.3 °C, respectively. Figure 4 presents bainite of the experimental steel.

Bainite is a non-lamellar structure composed of ferrite (α_B_) and carbides. Similar to this study, Rao et al. [29] observed island-like morphologies of bainite, with fine carbides dispersed on bainite. Bainite is an intermediate temperature transformation product obtained from the decomposition of supercooled austenite above the martensitic transformation temperature and below the pearlitic transformation temperature [30]. The formation mechanism of bainite is quite complex; researchers believe [31] that there is no diffusion of iron atoms in the bainitic transformation zone, and the growth of bainite is accomplished through the diffusion of carbon atoms. On the one hand, V can promote the formation of bainite and refine the microstructure. During plastic deformation, the concentration of plastic strain and the mobile dislocation density in bainite are lower than in ferrite [10]. As the volume fraction of the bainitic phase increased, so did the degree of strain gradient between the ferrite and the hard bainitic microstructure [32]. On the other hand, the formation of bainite can suppress the interphase precipitation of VC [29].

Figure 5a indicates that as the cooling rate increases from 1 °C·s^−1^ to 10 °C·s^−1^, the content of pearlite in the experimental steels initially increases and then decreases. The appearance of bainite at a cooling rate of 10 °C·s^−1^ leads to a reduction in pearlite content. Figure 5b shows that at the same cooling rate, the ILS for the three experimental steels follow the order LV > MV > HV, suggesting that increasing the V content can reduce the ILS. As the cooling rate increases from 1 °C·s^−1^ to 10 °C·s^−1^, the ILS of LV, MV, and HV steels decreases from 0.192, 0.180, and 0.166 μm to 0.117, 0.114, and 0.110 μm, respectively, showing reductions of 0.074, 0.065, and 0.055 μm. According to the metallographic classification standards [33], pearlitic steels can be subdivided into sorbite (0.08 μm < ILS < 0.15 μm) and troostite (0.03 μm < ILS < 0.08 μm). When the cooling rate is 5 °C·s^−1^, the ILS of HV steel first meets the criteria for sorbite. At a cooling rate of 10 °C·s^−1^, the microstructures of all three experimental steels are sorbite. Figure 6 presents the bright field and dark field TEM images of the phases of the experimental steels.

The morphology of cementite in pearlite can be continuous lamellae (as shown in Figure 6(a1,b1)), discontinuous lamellae, or granular (as shown in Figure 6(c1)). In Figure 6(c1), the growth direction of the cementite is more disordered, with horizontal, vertical, and oblique growth directions, and there is a significant variation in thickness between the cementites. The pearlite region containing such cementite can also be referred to as degenerated pearlite, which is caused by insufficient carbon content during the pearlite phase transformation [34,35]. The fine cementite particles exert a major dragging force on the migration of grain boundaries due to Zener pinning, which can lead to a very fine-grained microstructure [4]. TEM bright-field image analysis indicates that the thickness of the cementite ranges between 16.95 and 27.18 nm. When the cementite is subjected to high strain, cementites with a thickness of less than 200 nm are able to delay fracture through rotation and bending, thus ensuring a continuous obstruction to dislocation slip [36]. The green circular frames indicate that dislocations exist in both continuous and discrete pearlite. Studies have shown [37] that the smaller the spacing between cementites, the stronger the obstruction to dislocation slip. Dark-field image analysis indicates that there are a small number of round precipitates on the ferrite of the pearlite, with the size of the precipitates about 10–20 nm. When dislocation slip passes through the precipitates, the nano-sized precipitates are able to pin dislocations. The reduced quantity distribution of precipitate phases may be attributed to a higher cooling rate, which leads to a diminished contribution from precipitation hardening [10]. The research by Gu et al. [24] demonstrates that despite the nanoscale size of VC precipitates distributed around the cementite, their contribution to yield strength is minimal. This is because their localized distribution near existing cementite does not lead to a significant reduction in the average free path of dislocations. In this scenario, the nano-particles near cementite can be regarded as a single large precipitate with minimal contribution to the overall strength.

### 3.2. Precipitate

Precipitates can pin the migration of austenite grain boundaries, reducing the size of the prior austenite grains (PAGS) and ultimately decreasing the size of pearlite colonies [5]. Precipitates can also obstruct dislocation slip, enhancing the mechanical properties of steel. Figure 7 presents the TEM of VC precipitates.

Figure 7 indicates that the matrix is ferrite (α-Fe), with an exposed crystal plane of (110), an interplanar spacing at 0.2157 nm, and a crystallographic axis of [100]. The composition of the precipitates is VC, with a short axis length of approximately 24.73 nm, an exposed crystal plane of (002), an interplanar spacing at 0.2053 nm, and a crystallographic axis of [100]. These fine nano-sized VC precipitates can synergize with high-density dislocations, providing a higher capacity for continuous work hardening [38]. Research indicates [16] that the precipitation mode of VC is related to the cooling rate; slow cooling rates result in random VC precipitates, while high cooling rates produce planar VC precipitates. Figure 7e shows a phase transition zone between α-Fe and VC, where the ordered arrangement of VC gradually transitions to the ordered arrangement of α-Fe, with VC atoms gradually adjusting their positions to grow in the direction of α-Fe. Liang et al. [20] also observed a similar phenomenon in ferritic steel. Based on the Bramfitt two-dimensional misfit degree theory [39] (Equation (1)), a two-dimensional misfit degree model for α-Fe and VC was constructed (as shown in Figure 7f), and the misfit degree (δ) between α-Fe and VC was calculated. Table 2 is parameters of the two-dimensional misfit degree equation.
(1)$$\delta_{(hkl{)}_{n}}^{(hkl{)}_{s}}=\sum_{i=1}^{3}\frac{\frac{\left|(d[uvw{]}_{s}^{i}cos\theta )-d[uvw{]}_{n}^{i}\right|}{d[uvw{]}_{n}^{i}}}{3}\times 100$$
where (hkl)_s_ are the low-index crystal faces of the matrix, [uvw]_s_ are the low-index crystal directions on (hkl)_s_, (hkl)_n_ are the low-index crystal faces of the precipitates, [uvw]_n_ are the low-index crystal directions on (hkl)_n_, d[uvw]_s_ is the crystal face distance on [uvw]_s_, and d[uvw]_n_ is the crystal face distance on [uvw]_n_.

Based on Equation (1), the calculation yields δ = 5.02%, indicating the formation of a semi-coherent interface between α-Fe and VC. The (110) crystal plane of α-Fe is conducive to the nucleation of VC, and the semi-coherent interface can effectively reduce the interfacial free energy, ensuring the continuity of the ferritic matrix [40,41]. Furthermore, the fine VC precipitates can pin the α-Fe grain boundaries, reducing the migration rate of the boundaries and refining the grain size [22].

### 3.3. Effect of V Content on the Carbon Diffusion Coefficient

Pearlite is a diffusion-controlled phase transformation, with the growth behavior of pearlite being governed by the diffusion of carbon atoms. Fick’s law [42] indicates that the diffusion coefficient of carbon atoms determines their diffusion rate, with the formula expressing the carbon diffusion coefficient as a function of temperature.
(2)$$D_{C}^{0}=D_{0}exp(-\frac{Q_{i}}{RT})$$
where D_0_ represents the carbon diffusion coefficient in austenite, R is the gas constant, and T is the transformation temperature. The values of D_0_, Q_i_ and R in austenite are 0.23 cm^2^·s^−1^ and 138 kJ·mol^−1^ and 8.31 J·(mol·k)^−1^, respectively [43].

Lee et al. [44] proposed a quantitative relationship between the diffusion coefficient of carbon and the Nb content. Since both V and Nb are strong carbide-forming elements with similar chemical properties, they can both prevent the diffusion of carbon atoms. Therefore, Tian et al. [7] used the modified formula to reveal the influence of Nb and V on the diffusion coefficient of carbon atoms when they act together. Similar assumptions were made in this study when calculating the diffusion coefficient of carbon atoms.
(3)$$D_{0}=D_{C}^{0}\exp(-5000X_{V}(\frac{2750}{T}-1.85))$$
where X_V_ is the molar fraction of V, and $D_{C}^{0}$ represents the diffusion coefficient of carbon atoms without the addition of V.

Figure 8 shows the carbon diffusion coefficient for LV steel, MV steel, and HV steel.

Figure 8a shows that the carbon diffusion coefficients for three experimental steels are in the order LV > MV > HV, indicating that V in steel can drag the diffusion of carbon atoms, reducing the carbon diffusion coefficient. Figure 8b shows that as the cooling rate increases from 1 °C·s^−1^ to 10 °C·s^−1^, the carbon diffusion coefficients for LV, MV, and HV steels decrease from 0.13, 0.022, and 0.0034 cm^2^·s^−1^ to 0.046, 0.0076, and 0.98 × 10^−3^ cm^2^·s^−1^, respectively. The reduced carbon diffusion capacity in steel slows down the growth rate of pearlite, resulting in smaller pearlite colony sizes and ILS.

### 3.4. Effect of V Content on the Precipitation Behavior of VC

The precipitation of VC from steel requires a certain driving force, and can only occur when the free energy of phase transformation for VC is negative. Combining the regular solution sublattice model [45] with the solid solubility product equation [28], the phase transition free energy calculation formula for VC can be derived as follows:(4)$$∆G=-19.1446\times 9500+19.1446T(6.72-\mathrm{lg}\left(\left[V\right]\left[C\right]\right))$$
where [V] and [C] represent the concentrations of V and C in the steel, respectively.

The results of the phase transformation free energy calculation are shown in Figure 9.

Figure 9 indicates that the critical temperatures for the three experimental steels are in the order HV > MV > LV. This suggests that increasing the V content can raise the critical precipitation temperature of VC, promoting the precipitation of VC. Yong et al., based on nucleation growth theory and the Avrami equation, derived a precipitation kinetics model (Equations (5)–(7)) [28,46,47], and calculated the variation of VC precipitation time with temperature.
(5)$$\mathrm{lg}(\frac{t_{0.05a}}{t_{0a}})=n_{1}(-1.28994-2\mathrm{lg}d^{*}+\frac{1}{\ln10}\;\times \;\frac{\alpha \Delta G^{*}+n_{2}Q}{\mathrm{kT}})$$
(6)$$\alpha =A_{1}=\frac{1}{2}(2-3\cos\theta +\cos^{3}\theta )$$
(7)$$\alpha =(1+\beta {)}^{3/2}=(1+\frac{A\Delta G_{V}}{2\pi \sigma^{2}}{)}^{3/2}$$
where t_0a_ is the product of various parameters largely independent of temperature (based on the solute supersaturation function being essentially temperature-independent), t_0.05a_ is the time for the start of precipitation, defined as the fraction precipitated of 5% (s), t_0.05a_/t_0a_ is the relative time for precipitation, d* the critical nucleation radius, ΔG* is the activation energy for nucleation (J), k is the Boltzmann constant, and T is the temperature (K). When the precipitated phase is n_1_ = n_2_ = 1, α = A_1_ for grain boundary nucleation, and n_1_ = 2/3, n_2_ = 2.5, α = (1 + β)^3/2^ for dislocation nucleation [28]. A is the core energy per unit edge dislocation (J/m), ΔG_V_ is the volume-free energy (J/m^3^), σ is the interface energy between precipitates and matrix (J/m^2^), and Q is the atomic activation energy (J/mol).

Figure 10 shows that the nucleation rate order is dislocation lines > grain boundaries > homogeneous nucleation. As is well-known, the formation of VC precipitation nuclei requires a certain amount of energy to overcome the nucleation barrier. The disordered arrangement of atoms at grain boundaries and dislocations in steel both represent high-energy regions capable of providing energy for VC nucleation. Therefore, the temperature for heterogeneous nucleation (grain boundary and dislocation nucleation) is higher than that for homogeneous nucleation, and the time required for heterogeneous nucleation is shorter than that for homogeneous nucleation. Studies have shown that the atomic arrangement at grain boundaries is disordered, leading to the easy segregation and nucleation of V at grain boundaries [24,48]. The high density of dislocations between pearlite lamellae can capture free carbon atoms, making these sites preferential for VC nucleation [49,50]. With an increase in V content, the PTT curves shift to the left, and the critical precipitation temperatures for homogeneous nucleation, grain boundary nucleation, and dislocation line nucleation increase from 570.6, 676.9, and 692.4 °C to 634.6, 748.5, and 755.5 °C, respectively. Consistent with the phase transformation free energy results, this indicates that adding V can promote the precipitation of VC, increasing the VC content in steel and thereby improving the mechanical properties of the steel [24,26,50].

## 4. Conclusions

The microstructure, phase structure, and orientation relationship between α-Fe and VC of the experimental steels were characterized using SEM, XRD, and TEM. The proportions of microstructures and ILS were statistically analyzed, and the carbon diffusion coefficient, phase transformation free energy, and PTT curves of the experimental steels were calculated. The conclusions are as follows:The ILS for three experimental steels were in the order of LV > MV > HV. As the V content in the steels increased from 0.02 wt.% to 0.06 wt.%, the ILS of the experimental steels at four cooling rates (1, 2, 5, and 10 °C·s^−1^) decreased from 0.192, 0.178, 0.151, and 0.117 μm to 0.166, 0.157, 0.141, and 0.110 μm, respectively. At a cooling rate of 10 °C·s^−1^, a small amount of bainite is observed in the experimental steel, accompanied by a decrease in the content of pearlite.The carbon diffusion coefficients for three experimental steels were in the order of LV > MV > HV. As the V content increased from 0.02 wt.% to 0.06 wt.%, the carbon diffusion coefficients of the experimental steel at four cooling rates (1, 2, 5, and 10 °C·s^−1^) decreased from 0.138, 0.127, 0.103, and 0.0467 cm^2^·s^−1^ to 3.41 × 10^−3^, 2.93 × 10^−3^, 1.95 × 10^−3^, and 0.98 × 10^−3^ cm^2^·s^−1^, respectively.The precipitated phase was VC, with a diameter of approximately 24.73 nm. The misfit between α-Fe and VC was 5.02%, forming a semi-coherent interface between them. VC atoms gradually adjusted their positions to grow along the α-Fe direction. The ΔG of VC values for the three experimental steels were in the order HV > MV > LV. Within the same nucleation mechanism, the critical nucleation temperatures for the experimental steels are in the order HV > MV > LV. With an increase in V content, the PTT curves shift to the left, and the critical nucleation temperatures for homogeneous nucleation, grain boundary nucleation, and dislocation line nucleation increase from 570.6, 676.9, and 692.4 °C to 634.6, 748.5, and 755.5 °C, respectively.

## Figures and Tables

**Figure 1 materials-17-01392-f001:**
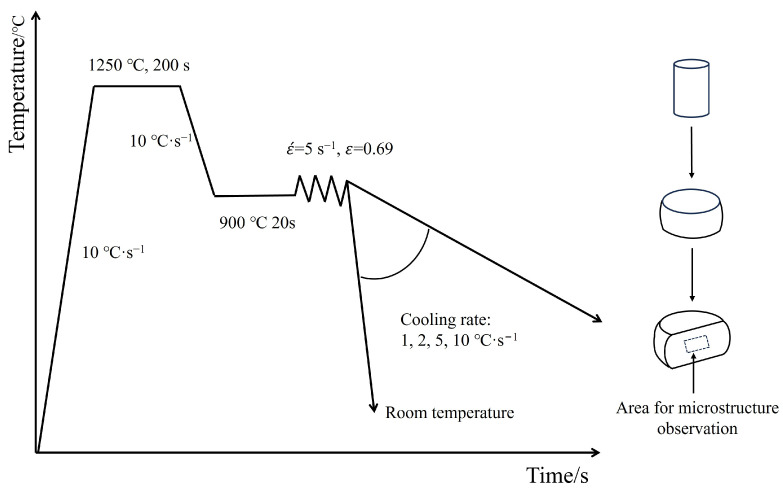
Thermo-mechanical simulation process of experimental steels.

**Figure 2 materials-17-01392-f002:**
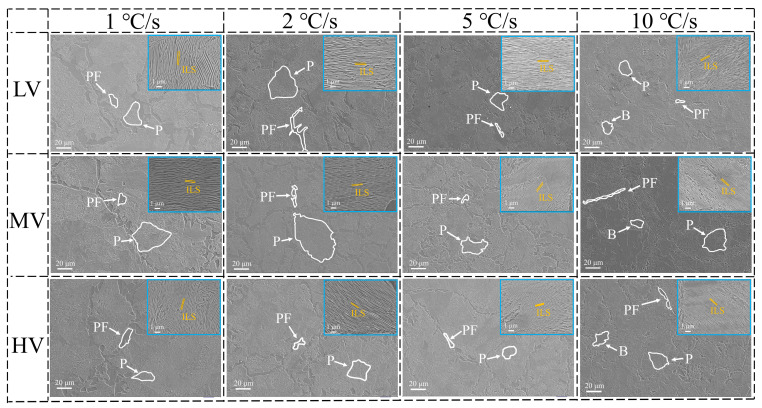
Microstructure and ILS of experimental steels at different cooling rates.

**Figure 3 materials-17-01392-f003:**
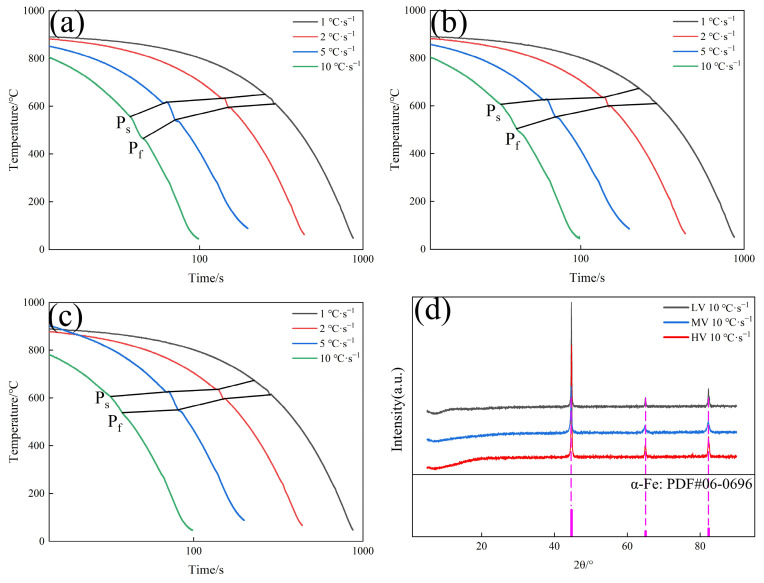
(**a**) CCT of LV steel; (**b**) CCT of MV steel; (**c**) CCT of HV steel; (**d**) XRD of experimental steels at 10 °C·s^−1^.

**Figure 4 materials-17-01392-f004:**
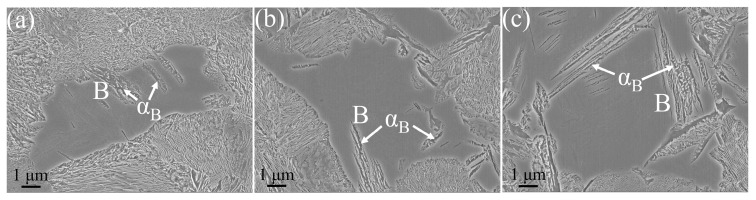
Bainite of the experimental steel: (**a**) LV steel, (**b**) MV steel, (**c**) HV steel.

**Figure 5 materials-17-01392-f005:**
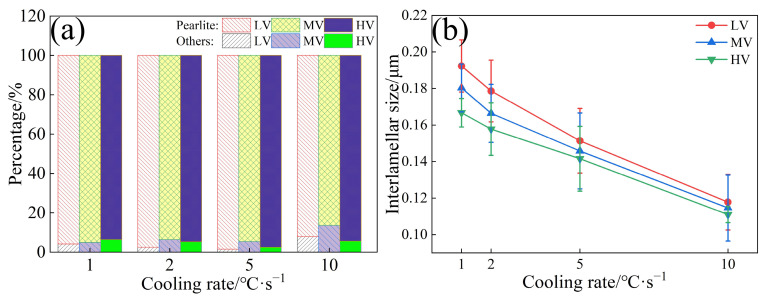
(**a**) Microstructural content; (**b**) ILS.

**Figure 6 materials-17-01392-f006:**
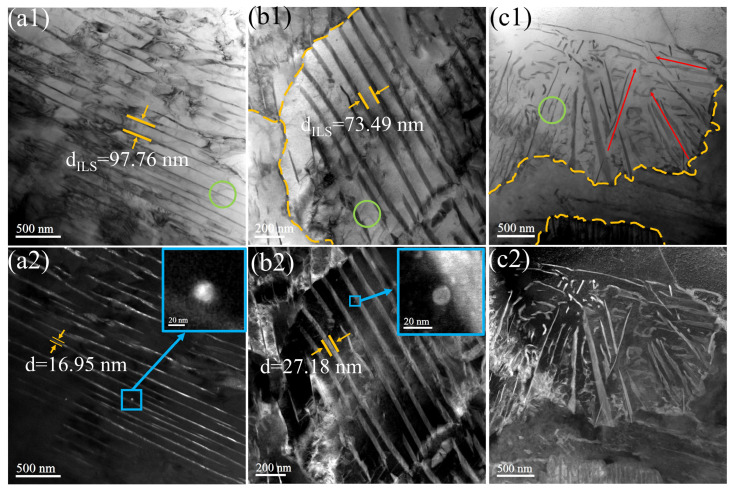
TEM of pearlite in experimental steels: (**a1**–**c1**) bright-field images; (**a2**–**c2**) dark-field images of the respective areas (green circular frames indicate dislocations, yellow dash lines are interfaces, red arrows indicate the growth direction of cementite, and blue frames indicate precipitates).

**Figure 7 materials-17-01392-f007:**
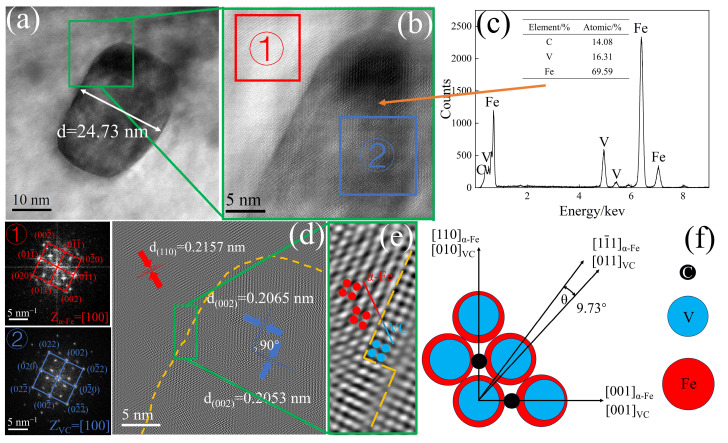
TEM of precipitates in experimental steels: (**a**) TEM of precipitates; (**b**) HRTEM of precipitates; (**c**) EDS of precipitates; (**d**) lattice fringes of precipitates; (**e**) magnified image of the green rectangular frame in (**d**); (**f**) two-dimensional misfit model of α-Fe and VC (① and ② are diffraction spots from area (**b**), yellow dash line is the interface, red and blue dots represent atoms).

**Figure 8 materials-17-01392-f008:**
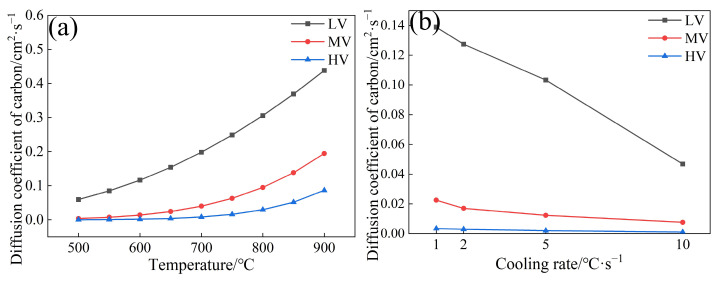
Carbon diffusion coefficient in experimental steels: (**a**) different V contents; (**b**) different cooling rates.

**Figure 9 materials-17-01392-f009:**
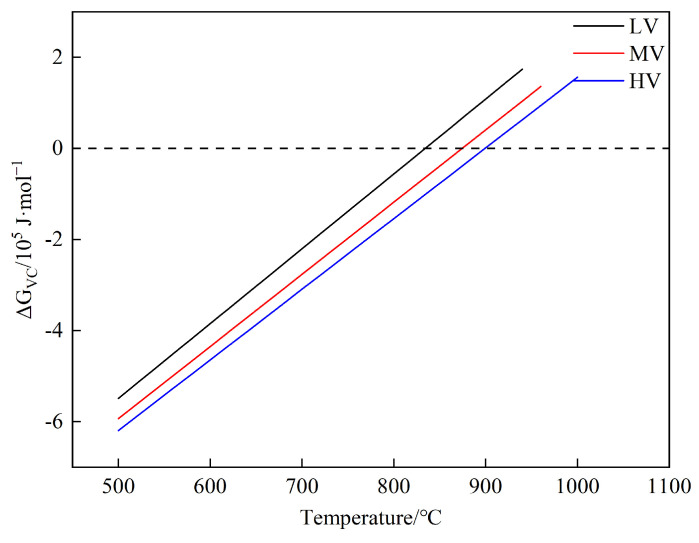
ΔG for precipitation of VC in three experimental steels.

**Figure 10 materials-17-01392-f010:**
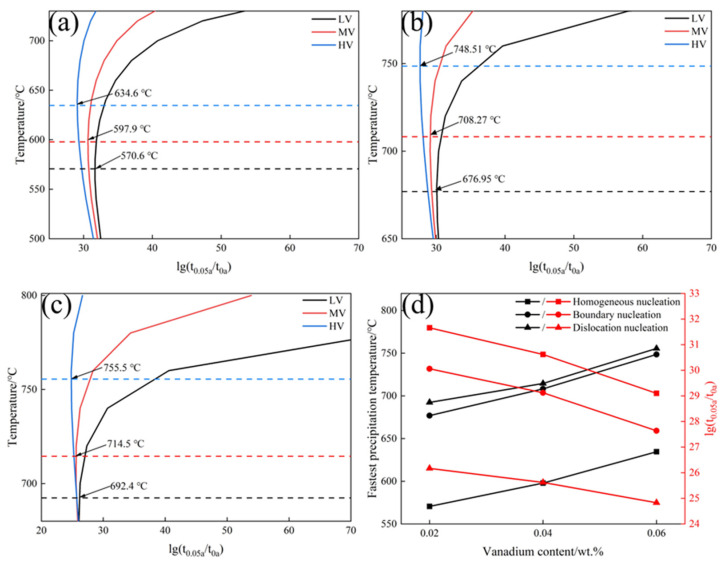
PTT curves of experimental steels with different V contents: (**a**) homogeneous nucleation; (**b**) grain boundary nucleation; (**c**) dislocation line nucleation; (**d**) critical precipitation temperatures and times for different nucleation mechanisms.

**Table 1 materials-17-01392-t001:** Chemical composition of experimental steels (wt.%).

Element	C	Si	Mn	P	S	Cr	Ni	Cu	V	Fe
LV	0.69	0.34	0.63	0.033	0.034	0.12	0.23	0.11	0.02	Bal.
MV	0.69	0.35	0.62	0.034	0.032	0.11	0.24	0.12	0.04	Bal.
HV	0.68	0.35	0.63	0.032	0.031	0.13	0.26	0.11	0.06	Bal.

**Table 2 materials-17-01392-t002:** Parameters of the two-dimensional misfit degree equation.

[hkl]_VC_	[hkl]_α-Fe_	d[hkl]_VC_	d[hkl]_α-Fe_	θ,deg	d[hkl]_VC_cosθ
[010]	[110]	4.162	4.145	0	4.162
[011]	$\left[1\bar{1}1\right)$	2.943	2.539	9.73	2.901
[001]	[001]	2.943	2.931	0	2.943

## Data Availability

Data are contained within the article.

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
