# Peer review of "Influence of V on the Microstructure and Precipitation Behavior of High-Carbon Hardline Steel during Continuous Cooling"

_materials, 2024, doi:10.3390/ma17061392_

Round 1
Reviewer 1 Report
Comments and Suggestions for Authors
Author Response
Thank you for your valuable feedback on the manuscript. We have made comprehensive revisions according to your suggestions and highlighted the modifications in red font within the manuscript. We have uploaded the revised manuscript along with point-by-point responses for your review.
Please see the attachment
Thank you and best regards!

Reviewer 2 Report
Comments and Suggestions for Authors
The manuscript titled "Influence of V on the Microstructural Transformation and Precipitation Behavior of High-Carbon Hardline Steel during Continuous Cooling" delves into the crucial role played by vanadium (V) microalloying in shaping the microstructure of high-carbon hardline steels. These steels hold immense significance in the fabrication of tire beads for automotive and aircraft applications. Through the meticulous use of scanning electron microscopy (SEM), X-ray diffraction (XRD), and transmission electron microscopy (TEM), the authors undertake a comprehensive investigation into the microstructural nuances and precipitation phases of continuously cooled high-carbon steels.
The experimental approach involves characterizing various aspects, such as vanadium content, carbon diffusion coefficient, and critical precipitation temperature. The findings highlight a direct correlation between the increasing V content (up to 0.06 wt.%) and the refinement of the interlamellar spacing (ILS) of pearlite, reaching an impressive 0.110 μm. Simultaneously, the carbon diffusion coefficient experiences a noticeable decrease to 0.98×10^-3 cm²·s^-1, indicating a transformative impact of V on the diffusion kinetics within the steel.
At specific cooling rates, the manuscript unveils intriguing trends in the pearlite content, with a maximum observed at 5 ℃·s^-1 for the experimental steel with 0.02 wt.% V. Furthermore, a shift to higher cooling rates reveals the emergence of bainite, adding a layer of complexity to the cooling rate-dependent microstructural evolution.
Intricate details regarding the precipitated phases shed light on the dominance of VC, characterized by a diameter of approximately 24.73 nm. The semi-coherent interface between ferrite and VC, with a misfit of 5.02%, underscores the nuanced atomic adjustments facilitating the growth of VC along the ferrite direction.
The manuscript also delves into the critical precipitation-temperature-time curve (PTT), revealing a leftward shift as V content increases to 0.06 wt.%. This shift significantly impacts the critical nucleation temperatures for homogeneous nucleation, grain boundary nucleation, and dislocation line nucleation, showcasing the intricate influence of V on the precipitation kinetics of high-carbon hardline steel.
In summary, the manuscript presents a thorough exploration of the influence of vanadium microalloying on the microstructural transformation and precipitation behavior of high-carbon hardline steel during continuous cooling. The insights provided contribute valuable knowledge to the materials engineering domain, with potential implications for optimizing mechanical properties in applications demanding high-performance steel. The experimental rigor and detailed analyses underscore the significance of this study within the broader context of alloy design and process optimization in steel manufacturing.
The following remarks should be adressed before the manuscript will be accepted:
- The novelty of the research should be clearly steated at the end of introduction section.
- The SEM images (Fig. 2) are taken at low magnification. It is very difficult to distinguish main microstructural features described in the article.
- Line 185 - "Figure 7 presents the TEM of VC. " - The text should reffer to Figure 6.
- Line 290 - "Authors should discuss the results and how they can be interpreted from the perspective of previous studies and of the working hypotheses. The findings and their impli- cations should be discussed in the broadest context possible. Future research directions may also be highlighted." - Why this text is in the discussion section ?
- Most of the references (more than 50% of them) are more than 5 years old.
Author Response
Thank you for your valuable feedback on the manuscript. We have made comprehensive revisions according to your suggestions and highlighted the modifications in yellow font within the manuscript. We have uploaded the revised manuscript along with point-by-point responses for your review.
Please see the attachment
Thank you and best regards!
